# Dose Contribution to the Regional Lymph-Node Metastases and Point B from Intracavity and Interstitial Hybrid Brachytherapy in Locally Advanced Cervical Cancer

**DOI:** 10.3390/cancers16132384

**Published:** 2024-06-28

**Authors:** Yoichi Muramoto, Naoya Murakami, Noriyuki Okonogi, Jun Takatsu, Kotaro Iijima, Tatsuya Inoue, Kanade Kato, Tatsuki Karino, Kanako Kojima, Masaki Oshima, Yasuo Kosugi, Terufumi Kawamoto, Takashi Hirayama, Kazunari Fujino, Yasuhisa Terao, Naoto Shikama

**Affiliations:** 1Department of Radiation Oncology, Juntendo University Graduate School of Medicine, 2-1-1 Hongo, Bunkyo-ku, Tokyo 113-8421, Japan; y.muramoto.kq@juntendo.ac.jp (Y.M.); t-kawamoto@juntendo.ac.jp (T.K.);; 2Department of Obstetrics and Gynecology, Faculty of Medicine, Juntendo University, 2-1-1 Hongo, Bunkyo-ku, Tokyo 113-8421, Japan

**Keywords:** intracavitary and interstitial hybrid brachytherapy, regional lymph-node metastases, point B, locally advanced cervical cancer, IGABT, optimal lymph-node boost dosage

## Abstract

**Simple Summary:**

The research investigates dose distributions to regional lymph-node metastases (RLNMs) in locally advanced cervical cancer (LACC) patients undergoing intracavitary and interstitial hybrid brachytherapy (IC/IS), a subject which has rarely been analyzed. Eleven sequential LACC patients with 38 RLNMs who received 38 IC/IS sessions were analyzed. RLNM dose distributions were assessed based on RLNM positions and ipsilateral interstitial needles, revealing significant differences in RLNM D_90_ based on whether the position was cranial or caudal of the uterine base and the number of interstitial needles was 0–1 or 2 or more. RLNMs located caudal of the uterine base with two or more ipsilateral interstitial needles received higher brachytherapy doses. The findings suggest that when calculating RLNM dose for external beam boost irradiation, consideration of RLNM position and interstitial needle count is crucial. This research sheds light on optimizing brachytherapy dosage delivery to RLNMs in LACC patients, potentially influencing treatment strategies.

**Abstract:**

Purpose: Analyzing dose distributions to regional lymph-node metastases (RLNMs) in locally advanced cervical cancer (LACC) patients undergoing intracavitary and interstitial hybrid brachytherapy (IC/IS). Methods: Dose distributions of eleven LACC patients with 38 RLNMs, and who received 38 IC/IS sessions were analyzed in EQD_2_, considering RLNM positions and ipsilateral interstitial needles; these RLNMs, excepting the para-aortic region, were classified into four groups. Results: RLNMs had a median of two ipsilateral interstitial needles per session. Significant differences were observed in total RLNM D_90_, depending on whether the position was cranial or caudal of the uterine base (85.5 vs. 378.9 cGy, *p* < 0.0001), and whether the RLNM D_90_ was associated with a number of ipsilateral interstitial needles between 0–1 or 2 or more (68.4 vs. 112.2 cGy, *p* = 0.006) per session. At each session, Group 1 RLNMs (cranial of the uterine base, 0–1 ipsilateral interstitial needle) had a mean D_90_ of 21.1 cGy; Group 2 (cranial, 2 or more), 73.8; Group 3 (caudal, 0–1), 94.7; and Group 4 (caudal, 2 or more), 136.1. Conclusion: RLNMs located caudal of the uterine base associated with two or more ipsilateral interstitial needles in IC/IS had a higher dose contribution, which should be considered when calculating the RLNMs’ dose of external beam boost irradiation.

## 1. Introduction

Management of locally advanced cervical cancer (LACC) requires a comprehensive therapeutic approach involving whole-pelvic external beam radiation therapy (WPRT) and brachytherapy [1]. The EMBRACE-I study demonstrated that image-guided adaptive brachytherapy (IGABT) can deliver >85 Gy to the high-risk clinical target volume D90 (CTV_HR_ D_90_) for more than 90% local control, regardless of T stage [2]. However, with only intracavitary brachytherapy (ICBT), this comes at the cost of late severe radiation toxicities, such as rectal bleeding, hematuria, rectovaginal fistula, and vesicovaginal fistula [3], especially for large and irregularly shaped tumors, because it is difficult to deliver the appropriate dose to the target while protecting the surrounding at-risk organs (OARs) within the dose restriction [4,5]. One of the good solutions to this problem is to perform intracavitary and interstitial hybrid brachytherapy (IC/IS), in which a few additional interstitial needles are inserted into areas difficult to cover with conventional ICBT in order to improve the dose coverage [6,7]. Due to its efficacy and affordability, IC/IS is associated with an expected faster implementation, compared to more complicated multi-catheter interstitial brachytherapy [8]. Another solution is inserting a spacer between the uterus/vagina and rectum/bladder to create a physical space and thus make it possible to reduce radiation exposure to the rectum and bladder [9,10,11].

On the other hand, control of regional lymph-node metastases (RLNMs) is an additional crucial aspect in the management of LACC. In brachytherapy, notably IC/IS, while primary tumor dose has been routinely evaluated, the dose contribution to RLNMs is generally not determined. Previous studies have demonstrated the significant contribution of image-guided adaptive high-dose-rate brachytherapy (IGABT) to RLNMs and point B, which is a reference point used in intracavitary brachytherapy since the 2-D era, and represents the dose contribution to the pelvic wall and is considered a surrogate for pelvic lymph-node doses in the form of an equivalent dose in 2-Gy fractions (EQD_2_) [12,13]. Different anatomical regional lymph-node groups, such as the para-aortic, common iliac, external iliac, internal iliac, pre-sacral, and obturator lymph nodes, have different dose contributions and require tailored treatment planning [14,15,16,17]. These investigations aimed to delve into how brachytherapy affects different RLNM groups in LACC. The dosage contribution of IGABT to pelvic lymph nodes has been studied [12,13,14,15,16,17], whereas IC/IS for LACC has not. IC/IS dose distribution can potentially extend widely in the lateral direction, compared to conventional ICBT, therefore potentially increasing the dose contribution to RLNMs. 

The purpose of this study is to provide a comprehensive evaluation of the dose distribution to RLNMs and point B during the IGBT era, focusing only on patients who underwent IC/IS, in order to determine the appropriate external beam boost radiation dose to the RLNMs for improving treatment precision and personalization.

## 2. Materials and Methods

In this study, patients with LACC and clinically identified regional lymph-node metastases, which were determined by PET-CT and MRI, and treated with definitive radiation therapy involving IC/IS were included. From 10 February 2023 to 30 September 2023, 11 consecutive LACC patients with 38 RLNMs were included. Table 1 summarizes patients’ backgrounds. The median age was 56 years (range: 30–78), and all patients had a Performance Status score of 0 to 1. The local stage was T2b:T3b in 5:6 patients, and all patients had lymph-node metastasis. The pathological findings reported squamous cell carcinoma in 9 patients and adenocarcinoma in 2 patients.

All patients received 45 Gy/25 fractions of WPRT (3DCRT in 9 patients and IMRT in 2 with para-aortic lymph-node metastases), without central shielding. Before taking the planning CT and for each external irradiation, bladder capacity was kept constant as to storage of urine by the drinking of 350 ml of water one hour before the CT scan or irradiation. Each PLNM received an additional 10.8–14.4 Gy/6–8 fractions (3DCRT in 4, IMRT in 7). Weekly cisplatin (wCDDP) 40 mg/m^2^ was administered concurrently for five cycles in seven patients, four in three, and three in one.

Depending on the extent of the parametrium invasion, IC/IS was carried out by inserting a few needles into the ipsilateral parametrium invasion transperineally or transvaginally, guided by transrectal ultrasound (TRUS) or transabdominal ultrasound [6]. A protocol was applied in which needles were inserted parallel, at intervals of 1.0–1.5 cm in cases of mass lateral invasion outside the reach of the tandem dosage. Without direct bladder or rectal invasion, a spacer (MucoUp®, Seikagaku Co., Tokyo, Japan) was inserted into the vesicovaginal septum and rectovaginal septum during each brachytherapy, as previously reported in detail [9]. Since all patients had LACC with parametrium invasion, IC/IS was performed in all cases. Six patients finished IC/IS in three sessions, while five had four. Overall, 38 IC/IS sessions were completed among the total number of patients. The median number of additional interstitial needles used for IC/IS was 2 (range; 1–6), with 0 (0–5) on the right side, 1 (0–5) on the left side, and 0 (0–1) on the dorsal side (behind the tandem for barrel-shaped disease).

According with the EMBRACE-I study, 80–85 Gy (EQD_2_) or higher was delivered to CTV_HR_ D_90_. Unlike EMBRACE-I, which contoured the CTV_HR_ using MRI, the CTV_HR_ was contoured using CT scans [18], suggesting that the CTV_HR_ could be overestimated, mostly in the lateral direction [19]. As a result, our CTV_HR_ D_90_ could be slightly lower than IGABT based on MRI. The total dose constraints of the organs at risk (OARs), combining WPRT and brachytherapy, were <75 Gy to the rectum, <90 Gy to the bladder, <75 Gy to the sigmoid colon, and <70 Gy to the small bowel in EQD_2_ (α/β = 3 Gy). 

Specific dosages of RLNMs and point B were evaluated at each IC/IS session. The doses to point B within the Manchester system at each IC/IS session were retrospectively examined, converted to EQD_2_, and totaled. Points B are defined as being two centimeters above the external cervical os of the uterus along the uterine axis, and five centimeters bilaterally from the midline [20].

Doses (D_100_, D_90_, D_50_, and D_0.1cm_^3^) for RLNMs were calculated at each session. The initial planning CT of the WPRT treatment was used to contour lymph-node metastases, which were then subdivided according to the definition of the lymph-node region. Calculations to sum the total dose of brachytherapy sessions were performed at OncentraBrachy (Elekta Solutions AB, Stockholm, Sweden).

We analyzed the statistical relationship between the dose contribution of each brachytherapy to point B and the RLNMs, the number of ipsilateral interstitial needles, and the location of lymph-node metastases.

Subsequently, we classified each lymph-node metastasis except the para-aortic region into four distinct groups, based on whether 0–1 or 2 or more ipsilateral interstitial needles were used at each IC/IS session and whether they were cranial or caudal of the uterine base. Each RLNM group D_90_ from IC/IS at each session was evaluated.

The Steel–Dwass multiple comparison test and the Wilcoxon signed-rank sum test were used to compare RLNM doses in terms of locations and ipsilateral interstitial needles. Spearman’s correlation coefficient was utilized to investigate the correlation between RLNMs and point B dosages; a significance level of *p* < 0.05 was established as the cut-off point, and JMP Pro 16 (SAS Institute Inc., Cary, NC, USA) was used.

The study was approved by the medical ethics management committee of our hospital (E23-0098) and carried out according to the principles of the Declaration of Helsinki.

## 3. Results

In EQD_2_, the median CTV_HR_ D_90_ in EQD_2_ for the 11 eligible patients was 85.4 Gy (range; 80.3–89.8), and in all patients achieved the OAR dose constraints: 65.5 Gy (53.4–69.8) to the rectum D_2cc_, 73.9 Gy (65.3–79.1) to the bladder D_2cc_, 57.2 Gy (42.7–68.9) to the sigmoid colon D_2cc_, and 47.2 Gy (45.9–58.6) to the small bowel D_2cc_.

Patients had a median of two regional lymph-node metastases (range; 1–11); the total number of RLNMs was 38. At the initial CT examination for WPRT, the median volume of RLNMs was 1.75 cm^3^ (0.4–14.1), with 21 (55.3%) cranial of the uterine base and 17 (44.7%) caudal. The detailed locations of the RLNMs and mean total RLNM doses from IC/IS are indicated in Table 2. The mean total D_90_ in EQD_2_ was 0.0 cGy for the para-aortic region, 48.5 cGy (range; 0.0–139.1) for the common iliac region, 313.8 cGy (95.8–578.9) for the external iliac region, 304.3 Gy (0.0–793.3) for the internal iliac region, 449.5 cGy (389.6–509.4) for the obturator region, and 480.3 cGy for the pre-sacral region. The total D_90_ of each lymph-node region differed significantly between the external iliac and the para-aortic regions (*p* = 0.039), and the external iliac and the common iliac regions (*p* = 0.014), but not between others. We also examined whether RLNMs in the cranial position relative to the uterine base increased the cumulative dose compared to the RLNMs in the caudal position relative to the uterine base. The mean total D_90_ of RLNMs in EQD_2_ differed significantly (*p* < 0.0001): 85.5 cGy (95%CI; 9.0–162.0) for cranial to the uterine base (*n* = 17), and 378.9 cGy (95%CI; 310.1–447.7) for caudal (*n* = 21).

Table 3 (*n* = 121, excluding para-aortic lymph-node metastases) shows the relationship between RLNM dose and the number of ipsilateral interstitial needles in each IC/IS session. There was a statistically significant difference in mean RLNM D_90_ in EQD_2_ at each session, ranging between 112.2 cGy (95%CI; 89.3–135.0) for 2 or more needles (*n* = 26) and 68.4 cGy (95%CI; 56.4–80.3) for 0–1 needle (*n* = 95) (*p* = 0.006).

Table 4 shows the mean total and each session’s dosage from IC/IS to point B. In EQD_2_, the mean bilateral point B dosage, which is the mean of both sides, was 185.4 cGy in each session and 640.6 cGy in total. The total D_90_ for the RLNMs (*n* = 34, excluding the para-aortic region) was correlated with the total dose at point B (Spearman’s correlation coefficient *ρ* = 0.59, *p* < 0.001). We also analyzed the correlation between the number of ipsilateral interstitial needles and each point B dose on the right and left sides (*n* = 76) in EQD_2_ at each session; this is summarized in Table 4: the mean point B dose of the ipsilateral zero needle (*n* = 15) was 144.8 cGy (range; 112–186), one needle (*n* = 46) was 173.7 cGy (103.8–280.6), two needles (*n* = 12) was 195.1 cGy (141.6–251.7), four needles (*n* = 1) was 283.2 cGy, and five needles (*n* = 2) was 652.3 cGy (567.7–736.9). The mean dose of point B in EQD_2_ was 166.6 cGy (95%CI; 156.8–176.4) for 0–1 ipsilateral needle (*n* = 61) and 262.0 cGy (95%CI; 170.0–353.9) for 2 or more needles (*n* = 15) at each session, a significant difference (*p* < 0.001).

Based on these results, we classified the IC/IS dose contribution to regional lymph nodes, excluding the para-aortic region, into four groups according to the following criteria: the location of lymph-node metastases (cranial or caudal to the uterine base), and the number of ipsilateral interstitial needles (zero or one, or two or more, at each IC/IS session). Consequently, the RLNM D_90_ in each group in each IC/IS session in EQD_2_ are calculated as follows: Group 1 (cranial and 0–1 needle, *n* = 34) was 21.1 cGy (95%CI; 7.7–34.4), Group 2 (cranial and 2 or more needles, *n* = 10) was 73.8 cGy (95%CI; 27.4–120.2), Group 3 (caudal and 0–1 needle, *n* = 61) was 94.7 cGy (95%CI; 82.7–106.7), and Group 4 (caudal and 2 or more needles, *n* = 16) was 136.1 cGy (95%CI; 104.7–167.6). There were significant differences between Groups 3 and 1 (*p* < 0.001), Groups 4 and 1 (*p* < 0.001), and Groups 2 and 1 (*p* = 0.008) (Figure 1).

## 4. Discussion

In this study, a comprehensive evaluation of the IC/IS dose distribution to RLNMs and point B for LACC patients was conducted, an important but often ignored aspect. While previous studies have examined the dose contribution of only IGABT, to the best of our knowledge, this is the first report analyzing the IC/IS dose contribution to point B and RLNMs comprehensively.

In 2009, Lee et al. reported results for point B dose and lymph-node dose in high-dose-rate IGABT, specifically, that the bilateral point B dose in EQD_2_ administered by IGABT was 141 cGy in each session and 7 Gy in a total of five fractions [12]. A single-center study from Japan also reported that the mean bilateral point B dose was 170 cGy per fraction with a 6 Gy prescription for high-dose-rate IGABT [21].

This study found the mean bilateral point B dose was 185.4 cGy in each session and 640.6 cGy in total from the IC/IS (Table 4). Although point B doses are in line with the results of previous high-dose-rate IGABT studies, our study showed that RLNM doses that were located closer to interstitial needles than to point B had a stronger influence from the IC/IS doses due to the intrinsic nature of the steep dose gradient.

While it would have been beneficial to explicitly compare the differences in dose distribution between intracavitary brachytherapy (ICRT) and IC/IS, it is important to note that all cervical cancers with lymph nodes were locally advanced and all underwent IC/IS procedures in this study. Consequently, we have elected to compare and analyze lymph-node metastases between the right and left side within the same patient where with and without an interstitial needle.

Even though previous studies that discussed point B doses showed a poor relation to the RLNM doses [12,13,15,21,22], there was a correlation between total RLNM D_90_ and total point B dose (Spearman’s correlation coefficient *ρ* = 0.59, *p* < 0.001), excluding para-aortic lymph nodes.

It was found that as the number of ipsilateral interstitial needles increased and the RLNM location was caudal to the uterine base, so the dosage of RLNM increased. The mean total RLNM D_90_ was significantly different between a number of ipsilateral interstitial needles ranging 0–1, and 2 or more needles (68.4 cGy vs. 112.2 cGy at each session), and between those located cranial and those located caudal to the uterine base (85.5 cGy vs. 378.9 cGy at total dosage).

Multiple studies suggest that 58 Gy or higher in EQD_2_ are preferred to control lymph-node metastases [23,24,25]. In the EMBRACE-I study, lymph-node metastases’ doses in the internal/ external iliac region were increased by up to 60 to 65 Gy [26]. In 2023, Brower et al. proposed a method for adjusting the recommended dose of external boost irradiation for RLNMs according to their size and location [14]. They sequentially raised lymph-node dosage to 55–59.4 Gy (57.2–61.8 Gy in EQD_2_) for obturator, external, internal iliac, and mesorectal lymph-node metastases, and 57.2–61.60 Gy (59.5–64.1 Gy in EQD_2_) for common iliac, presacral, and para-aortic lymph-node metastases.

Inspired by this research and aiming to simply understand the dose effects of IC/IS, we categorized RLNM treatments, excepting the para-aortic region, into four groups based on the RLNM location and the number of interstitial needles at each session (Figure 1). The mean D_90_ at each IC/IS session was 21.1 cGy (95%CI; 7.7–34.4) in Group 1 RLNM, which had 0 or 1 ipsilateral interstitial needle and was placed cranial of the uterine base, making it the least affected; 73.8 cGy (95%CI; 27.4–120.2) in Group 2 (2 or more ipsilateral interstitial needles and cranial); 94.7 cGy (95%CI; 82.7–106.7) in Group 3 (0–1 ipsilateral interstitial needle and caudal); and 136.1 cGy (95%CI; 104.7–167.6) in Group 4 (2 or more and caudal), the most impacted. And para-aortic lymph-node metastases were 0 cGy by IC/IS.

This simple classification may guide the potentially complicated dose calculation of the RLNM dosage in IC/IS and WPRT, as shown in Figure 2. Supposing that three to four IC/IS sessions would be performed and the mean D_90_ would be delivered in each Group (without changing the IC/IS method from beginning to end, including the number of additional interstitial needles), more than 57.2–57.4 Gy external beam boost irradiation would be required in Group 1 to achieve 58 Gy or more for RLMN control; more than 55.0–55.9 Gy in Group 2; more than 54.2–55.2 Gy in Group 3; more than 52.6–53.9 Gy in Group 4; and more than 58 Gy required in para-aortic lymph-node metastasis (Figure 2).

In practice, an accurate estimate of the number of interstitial needles needed for brachytherapy is required in order to adapt the lymph-node boost plan before IC/IS. Regarding the timeline of the decision-making process, pre-IC/IS MRI imaging should be used to decide on a lymph-node boost strategy by determining the number of interstitial needles required for IC/IS. Using the same number of interstitial needles in all IC/IS brachytherapy sessions would make it easier to estimate the total dose delivered from IC/IS brachytherapy to the lymph nodes. Assessing the lymph-node doses in each session of brachytherapy, at least for lymph nodes located caudal of the uterine base, will improve the accuracy of the RLNM dose evaluation. In the case where RLNMs are present in two or more groups simultaneously, using simultaneous integrated boost (SIB)-IMRT instead of 3DCRT treatment with lymph-node boost irradiation after 45 Gy WPRT would be an easier process.

This research has some limitations. First, for IC/IS, the number of interstitial needles and sessions varied slightly per session and patient. The same patient often required varied numbers of interstitial needles each session; thus, we could not divide the four groups to analyze the total dose of RLNM; instead, we grouped them according to the dose administered during each IC/IS session. Second, for the current investigation, we implemented a protocol by which needles were inserted at 1.0–1.5 cm intervals in cases of lateral mass invasion in regions unreachable by tandem dose. However, other institutions may observe different results if they follow a different policy, because the lateral pelvic wall dose can vary according to the method of brachytherapy dose calculation. More research is required to determine whether across multiple institutions the number of additional needles affects the dose administered to the lateral pelvic wall. Finally, further research is required, including prospective trials of the effects of the necessary dose for regional control on lymph-node metastases in IC/IS, given the limited number of cases included in this study and the oncologic outcomes not presented in this study.

The correlation between the control of regional lymph-node metastasis and overall survival has not yet been completely elucidated in cervical cancer. The rates of local recurrence, regional lymph-node recurrence, and distant recurrence in cases at our institution will be analyzed in the future, and the influence on overall survival will be clarified.

## 5. Conclusions

RLNMs located caudal of the uterine base, especially with two or more ipsilateral interstitial needles in IC/IS, receive a higher brachytherapy dose contribution.

## Figures and Tables

**Figure 1 cancers-16-02384-f001:**
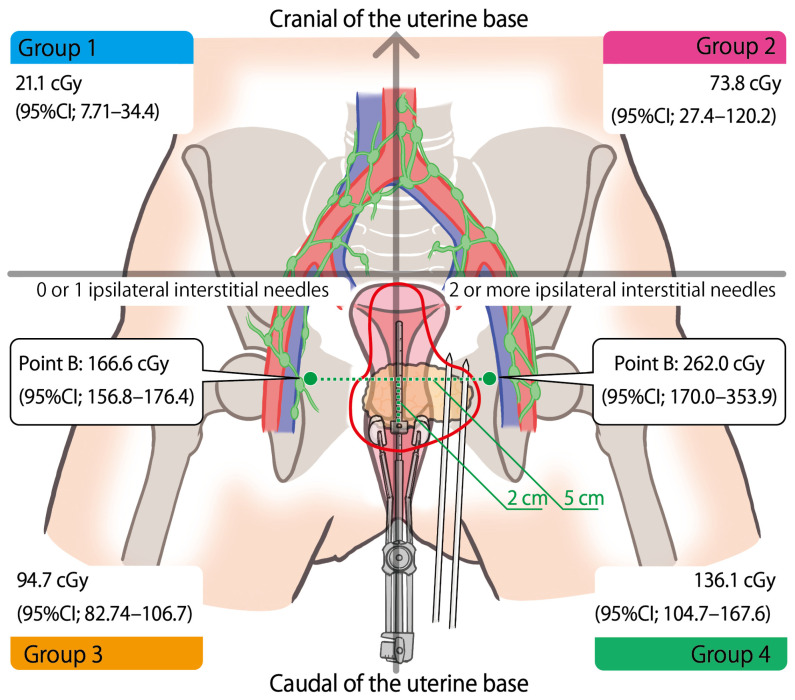
Regional lymph-node metastasis (RLNM) D_90_ of each Group and point B dose from IC/IS at each session in EQD_2_ are demonstrated. RLNMs, excluding the para-aortic region, are classified into four groups according to the location of lymph-node metastases (cranial or caudal to the uterine base), and the number of ipsilateral interstitial needles (zero or one, or two or more at each IC/IS session). Group 1 (left side, above) is RLNMs which have 0–1 ipsilateral interstitial needle and are cranial to the uterine base, Group 2 (right side, above) is 2 or more and cranial, Group 3 (left side, below) is 0–1 and caudal, Group 4 (right side, below) is 2 or more and caudal. Point B (green point) dose at each IC/IS session is also compared in categories of zero or one, or two or more ipsilateral interstitial needles. The emphasized red line represents the prescription isodose line, which is extended laterally with ipsilateral interstitial needles.

**Figure 2 cancers-16-02384-f002:**
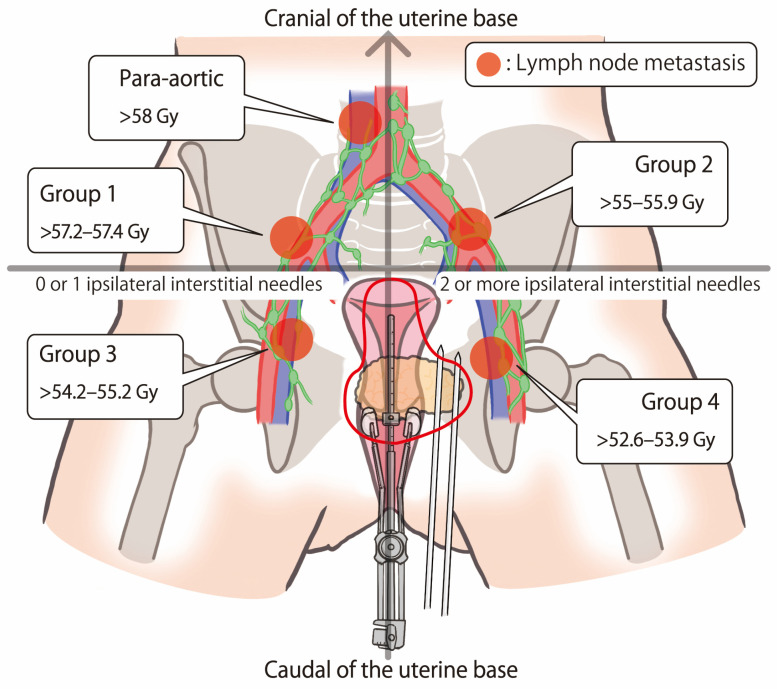
The proposed external beam irradiation prescribed doses required for lymph-node metastasis control to achieve over 58 Gy with IC/IS, in four groups and for the para-aortic region.

**Table 1 cancers-16-02384-t001:** Patients’ characteristics.

Variable		No. of Patients
Median age (years, range)	56 (30–78)	11
Median Body Mass Index (range)	23.2 (17.4–30.4)	11
ECOG PS (patients)	0	9
	1	2
T classification	T2b	5
	T3b	6
N classification	N0	0
	N1	11
FIGO stage (2018)	IIIC1	9
	IIIC2	2
Histology	Squamous cell carcinoma	9
	Adenocarcinoma	2

**Table 2 cancers-16-02384-t002:** Regional lymph-node metastases site doses and total doses from IC/IS in EQD2 (cGy).

Lymph Node Locations	Total Number (Left, Right Side)	Mean Total D_100_ (Range)	Mean Total D_90_ (Range)	Mean Total D_50_ (Range)	Mean Total D_0.1cm_^3^ (Range)	*p*-Value
Para-aortic lymph nodes	4 (2, 2)	0.0 (0.0–0.0)	0.0 (0.0–0.0)	0.0 (0.0–0.0)	0.0 (0.0–0.0)	-
Common iliac lymph nodes	6 (4, 2)	42.2 (0.0–119.7)	48.5 (0.0–139.1)	83.2 (0–219.1)	182.5 (75.8–410.0)	-
External iliac lymph nodes	14 (7, 7)	264.8 (72.1–516.5)	313.8 (95.8–578.9)	382.9 (109.8–690.9)	569.2 (137.3–1153.7)	-
Internal iliac lymph nodes	11 (6, 5)	264.4 (0.0–700.9)	304.3 (0.0–793.3)	379.0 (41.4–953.8)	578.6 (87.6–1412.5)	-
Obturator lymph nodes	2 (1, 1)	408.2 (351.3–465.1)	449.5 (389.6–509.4)	540.1 (494.1–586.1)	705.2 (660.7–749.6)	-
Pre-sacral lymph nodes	1 (0, 1)	451.2	480.3	526.9	633.6	-
Cranial of the uterine base	17	72.3 (0–386.3)	85.5 (0–473.0)	117.8 (0–602.8)	197.6 (0–870.8)	<0.0001
Caudal of the uterine base	21	328.9 (72.1–700.9)	378.9 (95.8–793.3)	460.0 (109.8–953.8)	672.1 (137.3–1412.5)	

IC/IS = intracavitary and interstitial brachytherapy, EQD_2_ = an equivalent dose in 2-Gy fractions, D_100_/D_90_/D_50_/D_0.1cm_^3^ = defined as the minimum dose covering 100%/90%/50%/0.1 cm^3^ of the volume of the regional pelvic lymph-node metastases (RLNMs).

**Table 3 cancers-16-02384-t003:** The mean regional lymph-node metastases doses per number of interstitial needles at each IC/IS session in EQD_2_ (cGy).

Number of Ipsilateral Interstitial Needles	Mean D_90_ at Each Session (Range)	*p*-Value
0 (*n* = 4)	48.6 (0–98.7)	-
1 (*n* = 91)	69.2 (0–285.8)	-
2 (*n* = 23)	106.2 (0–254.8)	-
4 (*n* = 1)	101.8	-
5 (*n* = 2)	185.6 (175.9–195.3)	-
0–1 ipsilateral interstitial needle (*n* = 95)	68.4 (95%CI; 56.4–80.3)	0.006
2 or more ipsilateral interstitial needles (*n* = 26)	112.2 (95%CI; 89.3–135.0)	

IC/IS = intracavitary and interstitial brachytherapy, EQD_2_ = an equivalent dose in 2-Gy fractions.

**Table 4 cancers-16-02384-t004:** The point B dose per number of interstitial needles at each IC/IS session in EQD_2_ (cGy).

Point B (11 Patients, 38 IC/IS Sessions)	Mean Dose of Each Session (Range)	Mean Total Dose (Range)	*p*-Value
Bilateral (mean of both side)	185.4 (106.9–484.5)	640.6 (455.9–1190.0)	-
Left side	203.0 (103.8–736.9)	703.0 (409.9–1587.8)	-
Right side	167.4 (109.9–280.6)	578.4 (377.2–792.1)	-
Ipsilateral 0 needle (*n* = 15)	144.8 (112.0–186.0)	-	-
Ipsilateral 1 needle (*n* = 46)	173.7 (103.8–280.6)	-	-
Ipsilateral 2 needles (*n* = 12)	195.1 (141.6–251.7)	-	-
Ipsilateral 4 needles (*n* = 1)	283.2	-	-
Ipsilateral 5 needles (*n* = 2)	652.3 (567.7–736.9)	-	-
0–1 ipsilateral interstitial needle (*n* = 61)	166.6 (95%CI; 156.8–176.4)	-	<0.001
2 or more ipsilateral interstitial needles (*n* = 15)	262.0 (95%CI; 170.0–353.9)	-	

IC/IS = intracavitary and interstitial brachytherapy, EQD_2_ = an equivalent dose in 2-Gy fractions.

## Data Availability

Data are contained within the article.

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
