# Peer review of "Dose Contribution to the Regional Lymph-Node Metastases and Point B from Intracavity and Interstitial Hybrid Brachytherapy in Locally Advanced Cervical Cancer"

_cancers, 2024, doi:10.3390/cancers16132384_

Round 1

Reviewer 1 Report

Comments and Suggestions for Authors

1.       Original Submission

Recommendation to the author and editor:

Major revision

Title: Manuscript ID: cancers-3022920 entitled "Dose Contribution to the Regional Lymph Node Metastases and Point B from Intracavity and Interstitial Hybrid Brachy-therapy in Locally Advanced Cervical Cancer.

Article Type: Original article

2.       Comments to the Corresponding Author:

COPE Ethical guidelines followed during the review process,

The manuscript addresses the dose distributions to regional lymph node metastases (RLNMs) in locally advanced cervical cancer (LACC) patients undergoing intracavitary and interstitial hybrid brachytherapy (IC/IS). Authors used the dose distributions of eleven LACC patients with RLNMs, who underwent 38 IC/IS sessions, were analyzed in EQD2. This analysis considered RLNM positions and the presence of ipsilateral interstitial needles. Authors explored that RLNMs, excluding those in the para-aortic region, were categorized into four groups based on their positions. Authors concluded that the RLNMs typically had a median of two ipsilateral interstitial needles per session. Significant variations were noted in the total RLNM D90 concerning whether the position was cranial or caudal to the uterine base (85.5 vs. 378.9 cGy, p<.0001). Additionally, RLNM D90 showed correlation with the number of ipsilateral interstitial needles, with differences between 0–1 needles and 2 or more (68.4 vs. 112.2 cGy, p=.006) per session. Authors concluded that RLNMs situated caudal to the uterine base with two or more ipsilateral interstitial needles during IC/IS demonstrated a higher dose contribution. Authors described that these factors warrants consideration in the calculation of RLNM dose for external beam boost irradiation. However, there are still some questions the authors need to answer.

Comments:

Overview and general recommendation:

The paper was well written. Yet, proofreading can enhance the quality of the manuscript. Several sentences need rewriting to make the readers comfortable when reading this. Avoid spelling errors.

1.               Authors should expand the discussion part with additional content.

2.     Interstitial brachytherapy (ISBT) is recommended for intact cervical carcinoma (IN-CC) when intracavitary brachytherapy (ICRT) is not viable. Although the patient pool for this study is limited, potentially leading to variations in dosimetric data. Authors can consider that the future investigations focusing on dosages to affected lymph nodes across various pelvic nodal groupings during ICBT could enhance our comprehension of the dosimetric landscape regarding inadvertent doses to implicated lymph node regions.

3.     What is the age range of the patients ?

4.     I am satisfied with the data pertinent to two or more ipsilateral interstitial needles in IC/IS exhibited a greater dose contribution for RLNMs. This factor merits consideration when determining the RLNMs dose for external beam boost irradiation. However, any impact of intracavitary and interstitial hybrid brachytherapy (IC/IS) on the patient’s overall survival. Can you elucidate the overall survival if there is any data by comparing the current dose distribution efficacy with other published reports to prove your hypothesis is more effective?

5.     Conclusion should be explained vividly

                                                **Thank you**

Comments on the Quality of English Language

Minor english language editing is required. 

Author Response

Reviewer #1: The paper was well written. Yet, proofreading can enhance the quality of the manuscript. Several sentences need rewriting to make the readers comfortable when reading this. Avoid spelling errors.

  • Authors should expand the discussion part with additional content.

Thank you very much for the insightful comments. According to the reviewer’s comments, the authors added the following sentences in the section “Discussion” as follows:

This study found the mean bilateral point B dose was 185.4 cGy in each session and 640.6 cGy in total from IC/IS (Table 4). Although point B doses are in line with the results of previous high-dose-rate IGABT studies, our study showed that RLNM doses that were located closer to interstitial needles than point B had a stronger influence from the IC/IS doses due to its intrinsic nature of steep dose gradient.

While it would have been beneficial to explicitly compare the differences in dose distribution between intracavitary brachytherapy (ICRT) and IC/IS, it is important to note that all cervical cancers with lymph nodes were locally advanced and all underwent IC/IS procedures in this study. Consequently, we have elected to compare and analyze lymph node metastases with and without an interstitial needle on the same side with lymph node metastases within the same patient.

・・・

Finally, further research is required, including prospective trials of the effects of the necessary dose for regional control on lymph node metastases in IC/IS, given the limited number of cases included in this study and the oncologic outcomes not presented in this study. The correlation between the control of regional lymph node metastasis and overall survival has not yet been completely elucidated. The rates of local recurrence, regional lymph node recurrence, and distant recurrence in cases at our institution will be analyzed in the future, and the influence on overall survival will be clarified.

  • Interstitial brachytherapy (ISBT) is recommended for intact cervical carcinoma (IN-CC) when intracavitary brachytherapy (ICRT) is not viable. Although the patient pool for this study is limited, potentially leading to variations in dosimetric data. Authors can consider that the future investigations focusing on dosages to affected lymph nodes across various pelvic nodal groupings during ICBT could enhance our comprehension of the dosimetric landscape regarding inadvertent doses to implicated lymph node regions.

Thank you very much for the insightful comments. As the reviewer pointed out, we would like to expand the number of patients in future studies to include patients treated with ICBT and would like to elucidate the differences in dosages for different pelvic nodal groupings during ICBT to further enhance our comprehension. Please let these topics be our future homework. The authors sincerely ask for the reviewer’s kind generosity regarding this issue.

  • What is the age range of the patients?

Thank you so much for pointing out an important point. According to the reviewer’s comments, the authors added the following sentences in the section “Materials and Methods” as follows:

Table 1 summarizes patients’ backgrounds. The median age was 56 years (range: 30-78), and all patients had a Performance Status score of 0 to 1. The local stage was T2b:T3b in 5:6 patients, and all patients had lymph node metastasis. The pathological findings were squamous cell carcinoma in 9 patients and adenocarcinoma in 2 patients.

  • I am satisfied with the data pertinent to two or more ipsilateral interstitial needles in IC/IS exhibited a greater dose contribution for RLNMs. This factor merits consideration when determining the RLNMs dose for external beam boost irradiation. However, any impact of intracavitary and interstitial hybrid brachytherapy (IC/IS) on the patient’s overall survival. Can you elucidate the overall survival if there is any data by comparing the current dose distribution efficacy with other published reports to prove your hypothesis is more effective?

Thank you very much for the insightful comments. According to the reviewer’s comments, the authors added the following sentences in the section “Discussion” as follows:

This study found the mean bilateral point B dose was 185.4 cGy in each session and 640.6 cGy in total from IC/IS (Table 4). Although point B doses are in line with the results of previous high-dose-rate IGABT studies, our study showed that RLNM doses that were located closer to interstitial needles than point B had a stronger influence from the IC/IS doses due to its intrinsic nature of steep dose gradient.

While it would have been beneficial to explicitly compare the differences in dose distribution between intracavitary brachytherapy (ICRT) and IC/IS, it is important to note that all cervical cancers with lymph nodes were locally advanced and all underwent IC/IS procedures in this study. Consequently, we have elected to compare and analyze lymph node metastases with and without an interstitial needle on the same side with lymph node metastases within the same patient.

・・・

Finally, further research is required, including prospective trials of the effects of the necessary dose for regional control on lymph node metastases in IC/IS, given the limited number of cases included in this study and the oncologic outcomes not presented in this study. The correlation between the control of regional lymph node metastasis and overall survival has not yet been completely elucidated. The rates of local recurrence, regional lymph node recurrence, and distant recurrence in cases at our institution will be analyzed in the future, and the influence on overall survival will be clarified.

5) Conclusion should be explained vividly

Thank you so much for pointing out an important point. According to the reviewer’s comments, the authors revised the following sentences in the section “Conclusion” as follows:

RLNMs located caudal of the uterine base, especially with two or more ipsilateral interstitial needles in IC/IS, receive a higher brachytherapy dose contribution.

The authors are appreciative of the reviewer’s perceptive comments on this article. It is our hope that the proposed revisions have effectively addressed the concerns raised by the reviewers. Should there be any further inquiries or issues that require clarification, we would be more than happy to provide additional information.

Reviewer 2 Report

Comments and Suggestions for Authors

This paper addresses a problem that is generally difficult to resolve in the case of cervical cancer, namely the involvement of the lymph nodes in the treatment. The presented conclusions offer a further element in this field

Author Response

Reviewer #2: This paper addresses a problem that is generally difficult to resolve in the case of cervical cancer, namely the involvement of the lymph nodes in the treatment. The presented conclusions offer a further element in this field

Thank you so much for grateful comments for us. We plan to continue our analysis of lymph node metastasis in cervical cancer.

The authors are appreciative of the reviewer’s perceptive comments on this article. It is our hope that the proposed revisions have effectively addressed the concerns raised by the reviewers. Should there be any further inquiries or issues that require clarification, we would be more than happy to provide additional information.

Reviewer 3 Report

Comments and Suggestions for Authors

First of all I have to congrats you for starting such researches which can help the other researchers and clinicians also physicist all around the world to compare new findings with the old ones and conclude better ways to treat the patients.

Also your used insertion techniques and delivered doses seems very conservative. I mean using just maximum 2 needles and the dose around 85 Gy.

By the way why did you use cGy instead of Gy somewhere? As a reader I prefer using the same unit at the whole article. Gy or cGy I mean, one of them.

The number of the patients also is not so high, but for starting it sounds good.

Author Response

Reviewer #3: First of all I have to congrats you for starting such researches which can help the other researchers and clinicians also physicist all around the world to compare new findings with the old ones and conclude better ways to treat the patients.

Also your used insertion techniques and delivered doses seems very conservative. I mean using just maximum 2 needles and the dose around 85 Gy.

By the way why did you use cGy instead of Gy somewhere? As a reader I prefer using the same unit at the whole article. Gy or cGy I mean, one of them.The number of the patients also is not so high, but for starting it sounds good.

Thank you very much for the insightful comments. We apologize for the inconsistency of the units regarding radiation dosages. To facilitate comprehension of the dosage scale, we employed Gy for external beam radiation and cGy for brachytherapy. We tried to establish a standard for cGy; however, the external beam radiation values would have been significantly higher, which would have impacted the charts. Consequently, we opted to maintain the current value. The authors sincerely ask for the reviewer’s kind generosity regarding this issue.

The authors are appreciative of the reviewer’s perceptive comments on this article. It is our hope that the proposed revisions have effectively addressed the concerns raised by the reviewers. Should there be any further inquiries or issues that require clarification, we would be more than happy to provide additional information.